# Effect of Sheep Grazing, Stocking Rates and Dolomitic Limestone Application on the Floristic Composition of a Permanent Dryland Pasture, in the Montado Agroforestry System of Southern Portugal

**DOI:** 10.3390/ani12192506

**Published:** 2022-09-20

**Authors:** Emanuel Carreira, João Serrano, Carlos J. Pinto Gomes, Shakib Shahidian, Luís L. Paniagua, Alexandre Pilirito, José Lopes Castro, Mário Carvalho, Alfredo F. Pereira

**Affiliations:** 1MED-Mediterranean Institute for Agriculture, Environment and Development, Instituto de Investigação and CHANGE—Global Change and Sustainability Institute, Institute for Advanced Studies and Research Universidade de Évora, Pólo da Mitra, Ap. 94, 7006-554 Evora, Portugal; 2Escuela de Ingenierías Agrarias, Universidad de Extremadura, Avenida Adolfo Suárez, S/N, 06007 Badajoz, Spain; 3Department of Animal Sciences, Universidade de Évora, Pólo da Mitra, Ap. 94, 7006-554 Evora, Portugal

**Keywords:** sheep, deferred grazing, continuous grazing, botanical composition, liming, dryland pasture, Montado

## Abstract

**Simple Summary:**

The Montado is a characteristic ecosystem of the Mediterranean region, where agricultural activities, animal production and forestry coexist alongside tourism, hunting and leisure activities. Animal grazing is fundamental for the conservation of the Montado, and it is imperative to clearly understand its interactions with the pasture floristic composition (PFC) of the Montado. The objective of this study was to evaluate the effect of sheep grazing, stocking rates and dolomitic limestone application on the floristic composition of permanent dryland pastures, in the Montado agroforestry system of Southern Portugal. The type of grazing influences the PFC, which may be positively or negatively impacted, depending on the adopted system. Deferred grazing seems to benefit the disappearance of undesirable plants and the appearance of desired plants. The results of this study allow for more informed management decisions and a potential increase in animal production but also improve the knowledge of conservation strategies in the Montado.

**Abstract:**

The Montado is a complex agroforestry–pastoral ecosystem due to the interactions between soil–pasture–trees–animals and climate. The typical Montado soil has an acidic pH and manganese toxicity, which affect the pasture’s productivity and pasture floristic composition (PFC). The PFC, on the other hand, can also be influenced by the type and intensity of grazing, which can lead to significant decreases in the amount of biomass produced and the biodiversity of species in the pasture. The objective of this study was to evaluate the effect of grazing type, by sheep, and different stocking rates on the PFC throughout the vegetative pasture cycle in areas with and without dolomitic limestone application. Thus, four treatments (P1UC to P4TC) were constituted: P1UC—without limestone application (U) and continuous grazing (CG); P2UD—U and deferred grazing (DG); P3TD—with the application of limestone (T) and DG; P4TC—T and CG. In DG plots, the placement and removal of the animals were carried out as a function of the average height of the pasture (placement—10 cm; removal—3 to 5 cm). The PFC was characterized in winter, at the peak of spring and in late spring. The PFC data were subjected to a multilevel pattern analysis (ISA). The combination of rainfall and temperature influenced the pasture growth rates and consequently the height of the pasture at different times of the year. Therefore, with the different growth rates of the pasture throughout the year, the sheep remain for different periods of time in the deferred grazing treatments. In the four treatments, 103 plant species were identified. The most representative botanical families in the four treatments were Asteraceae, Fabaceae and Poaceae. ISA identified 14 bioindicator species: eight for the winter period, three for the late spring vegetative period and three for the TC treatment.

## 1. Introduction

### 1.1. Overview of the Montado Ecosystem

The Montado is a multifunctional agro–silvo–pastoral ecosystem, characteristic of the Alentejo region (Southern Portugal). It is considered an ecosystem of “High Natural Value”, for the different productive and non-productive activities that it supports, as well as for being located in a region with low population density and scarce resources [1]. Agricultural, livestock and forestry activities are balanced in the Montado, as well as activities related to tourism and leisure, hunting, beekeeping, mushrooms and cork [1]. Thus, the Montado is associated with high complexity [2]. This complexity results from the interactions between the Mediterranean climate and the four fundamental components of the Montado: soil, pasture, trees and animals [2,3]. This complexity increases further due to the diversity of plant species in the pasture [4]. Most soils where the Montado is located are stony, acidic, poor in nutrients and suffer from nutritional imbalances, especially in terms of the magnesium/manganese ratio [2]. The Alentejo region, where the Montado is located, has a Mediterranean type of climate. This climate type is characterized by hot, dry summers and rainy winters, with mild temperatures [5], significant seasonality and variability [3]. Prolonged natural droughts often impair pasture production. Moreover, the precipitation variability, either in quantity or in seasons, affects pasture productivity and quality [6]. The spontaneous pastures of the Montado ecosystem generally have low productivity [7]. One of the agronomic techniques to improve this natural pasture’s productivity involves applying phosphate fertilizers [8] and correcting manganese toxicity [9], through the application of dolomitic limestone. On the other hand, the low yield of animal production is associated with extensive production systems. Consequently, low investment in these systems leads to little knowledge of the relationship between the effects of different types of grazing and pasture productivity [10]. Thus, it is crucial to carry out different trials to better understand the impact of limestone and stocking rate on the biodiversity of the pasture, the evolution of the plant species and the existing families [4].

### 1.2. Effects and Relationship of Different Grazing Systems on Pasture Floristic Composition

The grazing system and the way in which it is managed can determine the pasture floristic composition (PFC), even in overseeded pastures [11], where, for example, in a grass pasture with white clover (50/50), the percentage of white clover can vary from 1 to 80% after a few years, depending on the number of weeks between each grazing event: if grazing is carried out every week, its percentage is 80%; if grazing takes place every 4 weeks, its percentage is 50%; if it takes place only every 12 weeks, its percentage is only 1% [12]. Plant community compositions are affected by selective grazing, stocking rate and grazing seasons [13]. In a livestock system with multiple species, there is a tendency towards selectivity in the consumption of the same species, which varies according to the phenological stages of the different species throughout the year. Increasing the instantaneous stocking rates can help to reduce selectivity and thus avoid the overgrazing of more edible species and undergrazing of less palatable species, preventing them from becoming dominant in the pasture [14]. Moreover, the rest periods of grazing are essential for the plants to develop, become vigorous and produce seeds. This is most beneficial for the more palatable species, and results in the high production of grasses [15].

Currently, the most common grazing systems are continuous grazing (CG) and rotational grazing (RG) [16]. We refer to CG and deferred grazing (DG) in the present study. DG, in this case, is associated with longer or shorter grazing periods, with instantaneous stocking rates, depending on the pasture’s quantity. In rotational grazing, the animals remain for a fixed period in each pasture plot and there is an absence period that depends on the number of plots. In deferred grazing, the stays differ depending on the pasture’s biomass and the exclusion criteria (which can vary depending on the species and their growth habits).

The main problem of CG is the selectivity displayed by the animals, which results in areas that are heavily grazed and others that are not grazed [15,17]. However, animals select different plants and parts of the plants depending on the season [15]. DG allows plants in preferred areas to grow and recover [17], which would not be possible under CG. In New Zealand, DG was applied successfully to improve productivity, resilience and pasture recovery [14]. DG leads to pasture improvement, through increased dry matter (DM) production, ryegrass percentage and soil cover, without negative impacts on pasture quality after the removal of the animals [18]. On the other hand, when we increase the grazing pressure (higher stocking rate), we delay the vegetative cycle of the plants, providing a greater number of green leaves and, consequently, an increase in the quality of the pasture [19]. Mccallum et al. [20] report that pastures grazed under DG produced an additional 2.7 ton DM/ha when compared to pastures under RG. In a research work, which compared CG with a low biotic load and RG with high biotic loads, the results showed that pasture production is higher when the animal load is higher [16]. Moreover, Brougham [11] mentioned that DM production is higher in a grazing system with higher biotic loads in winter than in grazing with low biotic loads. However, according to Heady [15], in grazing systems where animals have more difficulty choosing their diet, as is the case of DG, by forcing animals to consume diets of better nutritional value, we can also improve their productive performance.

PFC is a good field indicator of biodiversity as well as pasture quality. Pastures composed of multiple species are more resilient to the climatic variations that are so common and may present advantages in the complementary growth that they present, enabling biomass with acceptable nutritional value for the animals [21]. To change the PFC of different pasture plant communities, we need to understand the effects of grazing management on the restoration of seedlings [22]. The grazing system chosen by the livestock producer affects the PFC and the performance of the different plant species [11]. Even if a pasture is overseeded with a mixture of high-quality seeds, if the pasture is poorly managed, it quickly turns into a degraded pasture with many unwanted plants for animals [12]. Pasture degradation leads to a decrease in biomass productivity and increased risk of erosion by wind and rain [23]. Grazing with sheep, with high stocking rates, can harm pastures, leading to a reduction in the diversity of species [24]. However, this is not always the case; it is necessary to carry out studies to understand better the interactions among the type of animals, the type of grazing, stocking rates, season, duration and initial PFC [24]. This study is one of the first to assess the effects of grazing type and the application of dolomitic lime to the soil on the evolution of the PFC throughout the year, under the Montado ecosystem. On the other hand, adequate pasture management makes it possible to recover degraded areas, in good-quality pastures. Although CG and RG are managed differently, even if the stocking rates are similar, the effects on the pasture will be different [25]. In regions where the climate is irregular, as is the case of the Alentejo region, it is not possible to improve the plant communities simply by removing the animals in specific periods (DG), since the response of plants is rather unpredictable [26,27] and dependent on precipitation distribution and temperature. For DG to contribute to the improvement of the PFC, the amount of desirable plant species should be at least 20%, and sufficient livestock should be available to graze adequately and quickly at the right time [28], so that there is a similar removal of biomass throughout the plot, without any preferred areas. DG is a flexible and inexpensive technique that improves pastures [20]. Nevertheless, according to Edwards et al. [25], the survival of seedlings of some edible good species (such as *Lolium perenne* L. and *Trifolium repens* L.) and less edible species such as *Cirsium vulgare* (Savi) Ten., *Rumex obtusifolius* L. and *Plantago lanceolata* L. in winter was higher in CG treatments than in RG treatments.

Considering that, during pasture regrowth, in RG systems, the pasture changes in biomass height and soil cover, it can be inferred that this grazing system is more favorable to the emergence of seedlings, when compared to the CG system [25]. 

However, according to Voisin and Lecomte [12], DG is the best technique to recover degraded pastures and improve their PFC, leading to an increase in the percentage of legumes, namely white clover. In DG systems, in order to improve the PFC, in early spring, before the production of the inflorescence of grasses, it is crucial to carry out grazing with a high animal load [28], to try to lengthen the vegetative cycle of the pasture. 

Animal production, based on grazing, contributes to the maintenance and improvement of soil fertility, reducing animal feed costs [29]. Furthermore, it is essential to develop grazing systems that reconcile the need for agricultural productivity with environmental aspects [23].

This study aimed to evaluate the effect of the type of sheep grazing (continuous vs. deferred) with different stocking rates on the floristic composition of permanent dryland pastures in the Southern Alentejo region. This evaluation was performed in areas with and without the application of dolomitic limestone in winter, at the peak of spring and in late spring. 

## 2. Materials and Methods

### 2.1. Study Area Framework

This study sequences other trials conducted from 2015 to monitor the effect of dolomitic limestone application on soil, tree, pasture and sheep grazing interactions over time (Figure 1), which resulted in some scientific articles [4,30,31,32,33]. 

The predominant soils of this region are classified as Cambisol, derived from granite, which commonly has low fertility [34]. The study area is in a large patch of holm oak (*Quercus rotundifolia* Lam.), with an average density of 9–10 trees per hectare [35], over an understory of dryland pastures, mostly used for extensive animal production, especially to produce beef cattle and sheep. The Alentejo is affected by the Mediterranean climate. This climate is characterized by hot and dry summers, with maximum temperatures above 40 °C, and wet and cold winters, with minimum temperatures below 0 °C [32,33]. The irregular rain distribution and total year precipitation variation are also characteristic of the Mediterranean climate. In this region, the total amount of annual precipitation varies from 300 mm to 650 mm [33], with most of this precipitation occurring in autumn, winter and spring. In summer, if there is any precipitation, it will always have residual values.

The present study was carried out between November 2020 and June 2021. In this region, there is a large area of the Montado, mostly used for extensive cattle and sheep production systems. 

### 2.2. Study Design Description

The study took place in an area of approximately 4 ha, subdivided into 4 plots of 1 hectare each (Figure 2) (38°32.2′ N; 8°1.1′ W), located in the Mitra farm in the Alentejo region, Portugal.

The characterization of the surface layer of the soil (0–0.30 m depth), carried out in October 2015, revealed an acidic pH (average value of 5.4±0.3), so two amendments with lime were carried out (2 ton/ha of dolomitic limestone) in half the area (P3TD and P4TC) in November 2017 and June 2019. In December 2018, the whole study area (P1UC, P2UD, P3TD and P4TC) received 100 kg/ha of binary fertilizer (18-46-0). The experimental design was based on a factorial scheme, with two plots subjected to the application of dolomitic limestone and two others serving as controls (UC treatments). Within each treatment with and without amendment with dolomitic limestone, two grazing systems were applied: CG with continuous grazing and a moderate stocking rate and DG with deferred grazing and a high stocking rate (2 times that applied in the continuous grazing scenario). The four treatments were as follows: Plot 1 (P1UC)—without dolomitic limestone application and CG (7 sheep/ha); Plot 2 (P2UD)—without dolomitic limestone application and DG (16 sheep/ha); Plot 3 (P3TD)—with dolomitic limestone application and DG (16 sheep/ha); Plot 4 (P4TC) (7 sheep/ha)– with dolomitic limestone application and CG. 

### 2.3. Grazing Management

The project that allowed the development of this study began in 2018, with the collection of elements regarding soil evolution and the influence of trees on the pasture’s growth and nutritional value [4,31,32,33]. During this period, this pasture was grazed by the same herd that were studied in 2020 and 2021. The grazing was carried out with non-pregnant or lactating adult White Merino and Black Merino ewes (Figure 3). All ewes had similar body conditions at the beginning of the trial. All animals had a mean body condition score (BCS) of 3.5, with a standard deviation of 0.5. The scale used is from 1 to 5, where 1 is very thin and 5 is obese [36].

The sheep in the P2UD and P3TD plots started grazing at the beginning of the experimental period, but according to a DG system. The presence or absence of animals was linked to pasture conditions following the “put and take” method used by [37,38]. Grazing management criterion was a function of the average pasture height in each plot, measured with a precision digital caliper. When the pasture’s average height was less than 3 to 5 cm, the animals were removed and placed in an annex plot outside the study area (Figure 2), where they were fed until the pasture recovered and reached a mean height of 10 cm. 

Pasture heights were measured in the 4 plots before and after each grazing period. Pasture samples were collected to estimate the productivity of green matter (GM) and dry matter (DM), both in Kg ha^−1^. At the same time, the crude protein CP and neutral detergent fiber, NDF, were evaluated based on the methodology proposed by [39,40]. Every month, all the animals were evaluated in terms of their body condition to highlight possible weight loss or variations among the animals’ body conditions [41], in the different plots.

### 2.4. Characterization of the Floristic Composition

Forty-eight sampling points were chosen to identify variations throughout the year in the relative proportions of the different species, 12 in each treatment (Figure 4). Each sampling point was permanently marked with a numbered flag (1 to 12 in each plot) (Figure 4). 

Each of these 12 points represents, in each plot, a plant community, with species that vary in diversity and occurrence. The characterization of the floristic composition was carried out on January 14th (winter—WI), May 4th (peak of spring—SP1) and June 17th (late spring—SP2) of 2021. This characterization involved the identification of different plant species on each date in an area of 1 m^2^. For each species, presence or absence was noted in each of the 12 points. The relative abundance of the various species present was also measured. However, the results will be presented in the following article, where the behavior and food preferences of the sheep will be analyzed.

### 2.5. Statistics Analyses

Data were first organized and processed in a spreadsheet for descriptive analysis. In addition, species were organized by family and by occurrence vs. absence in each study plot.

Subsequently, the data were subjected to a statistical analysis, namely multilevel pattern analysis (Indicator Species Analysis—ISA), a specific package in the “R” statistic software (St. Louis, MO, USA) [42]. The ISA involves the calculation of an indicator value (IV) for plant species, corresponding to the product between relative abundance (specificity) and relative frequency (fidelity) expressed in degrees (in percentage) [43]. However, as our data were merely the presence/absence of species and not the percentage of each species relative to others, the data had to be transformed by the Beals Smoothing transformation method [44], aiming to understand whether the treatments (CG vs. DG and limestone application vs. no application) impacted pasture biodiversity (i.e., the number of species present), rather than quantifying the percentages of each species. This team has already used this approach to quantify the percentages for each species in other published works. It requires exhaustive, time-consuming monitoring, which is incompatible with the demand for quick responses at the scale of plots corresponding to large areas.

To reduce the problem of data analysis in which we only have information regarding the presence (1) or absence (0) of species, the "sociological favorability index" (SFI) was used [44]. This index assesses the probability of occurrence of each species in each location based on their joint occurrence with other species [45]. With this transformation, each cell value (1 or 0) was replaced by the occurrence probability of each species in each sample unit. A bioindicator species was carried out based on time (1- WI, 2- SP1, 3- SP2) and treatment (T1=P1UC, T2=P2UD, T3=P3TD, T4=P4TC). A significance level (α= 5%) was used.

## 3. Results

### 3.1. Meteorological Conditions

Figure 5 represents the thermopluviometric graph for Mitra, between September 2020 and June 2021. The total amount of precipitation in this period was 627.8 mm, distributed very erratically over the various months, affecting the growth and development of the pasture. As shown in Figure 5, in September, the 40 mm of precipitation, together with the 133 mm in October, provided the moisture necessary for the germination and growth of the pasture. In March, precipitation was almost absent, with a residual value of 12 mm. In addition, the spring of 2021 was quite arid, with 7.7 mm and 10.4 mm of rainfall in May and June, respectively. In Figure 5, the grey line represents the monthly average maximum temperature, the orange line represents the monthly average minimum temperature, and the yellow line represents the monthly mean temperature. It is worth highlighting the temperature values for September and October, with a monthly average of 22.4 °C and 16.3 °C, respectively. The lowest temperatures and, therefore, the most limiting period for pasture growth occurred in January, with a minimum average of 3.4 °C. During this month, the average monthly temperature was 8.1 °C, and the average maximum was 13.7 °C. In this spring period, the average maximum temperature was 25.9 °C and 29.9 °C for May and June, respectively. The average minimum temperature was 9.9 °C and 12.5 °C for May and June, respectively. As we can see in Figure 5, the lowest temperatures occurred in winter, when there was greater water availability in the soil. On the other hand, in the spring months, water availability in the soil was relatively limited due to low precipitation in this period. 

### 3.2. Grazing Days

Figure 6 shows the number of grazing days in each plot over the months, and Figure 7 shows the total grazing days. In December, in the P2UD plot, the animals only grazed until the 11th and in the P3TD plot until the 17th. We must highlight here the month of January, where, in the plots designated as DG (P2UD and P3TD), the animals were not present during the whole month. Moreover, in February and March, the plots intended for DG were left vacant during roughly half of each month, so that the pasture could recover. In February and March, the numbers of grazing days for P2UD and P3TD were 17 and 14, respectively, for each month. In May and June, in all plots, the grazing days were the same. In the month of April, at P2UD and P3TD, the animals were only out for 8 days. In P1UC and P4TC, the total grazing days were 236. In the P2UD plot, the animals grazed for 151 days, and in the P3TD plot, the grazing days were 158. In other words, P3TD had 7 more grazing days than P2UD. 

### 3.3. Characterization of the Floristic Composition

#### 3.3.1. Descriptive Analysis

In total, in WI, SP1 and SP2, 103 plant species were identified, belonging to 25 families.

The plant species that were identified in this study in each plot and in WI are shown in Table A1 (Appendix A). A total of 51 different species were identified, belonging to 15 botanical families. The most common species, in all plots, was *Vulpia geniculate* L. Other species, such as *Bromus diandrus* Roth, *Diplotaxis catholica* (L.) DC., *Echium plantagineum* L., *Erodium cicutarium* subsp. *bipinnatum* (Cav.) Tourlet, *Geranium molle* L. or *Leontodon taraxacoides* (Vill.) Mérat and *Senesio vulgaris* L., were also identified in all plots in WI, at many of the sampling points. 

Table A2 (Appendix A) shows the plant species that were identified in each plot, in SP1. A total of 78 species were identified, belonging to 23 botanical families. In SP1, *Bromus diandrus* was not identified at any sampling point. However, the number of species with a more significant presence was higher in SP1 than in WI. The following can be noted: *Bromus hordeaceus* L., *Chamaemelum mixtum* L., *Crepis capillaris* (L.) Wallr., *Diplotaxis catholica*, *Echium plantagineum*, *Erodium cicutarium* subsp. *bipinnatum*, *Geranium molle*, *Hedypnois cretica* (L.) Dum.-Courset, *Hordeum murinum* subsp. *leporinum* (Link) Arcang., *Plantago coronopus* L., *Plantago lagopus* L., *Rumex bucephalophorus* L., *Tolpis umbellata* Bertol., *Trifolium campestre* Schreb., *Trifolium glomeratum* L. and *Vulpia geniculata* (most numerous).

The plant species identified in the four plots in SP2 are indicated in Table A3 (Appendix A). On this date, 53 species belonging to 17 families were identified. The most prominent species continued to be *Vulpia geniculata*. The following species are also highlighted: *Agrostis pourretii* Willd., *Chamaemelum mixtum*, *Crepis capillaris*, *Echium plantagineum*, *Hordeum murinum* subsp. *leporinum*, *Plantago lagopus* and *Tolpis umbellata*.

Figure 8a represents the number of plant species per family observed in WI, SP1 and SP2 at P1UC. In this plot, 76 different plant species were identified. In the plot, three families had the highest number of species in the three seasons: Asteraceae, Fabaceae and Poaceae. In the Asteraceae family, the most significant number of species occurred in WI, with 12 species, followed by 10 species in SP1 and 9 species in SP2. In the Fabaceae family, the most significant number of species occurred in SP1 (10 species), followed by SP2 (7 species) and WI (3 species). In the Poaceae family, eight species were identified in SP1 and SP2, and only four in WI. It should be noted that no plant species were identified in the P1UC belonging to the families Cucurbitaceae and Cyperaceae. In many other families, as shown in Figure 8a, only one or two species were identified in at least one season.

Moreover, in P2UD, the most numerous plant families were Asteraceae, Fabaceae and Poaceae (Figure 8b). The Asteraceae family comprised 10 species in WI, 9 in SP1 and 4 in SP2. The Fabaceae family was very numerous in SP1, with nine identified species, while only two and oone species were present in WI and SP2, respectively. The Poaceae family comprised six species in WI and eight in SP1 and SP2. As in the case of the P1UC and the P2UD plots, not all of the species identified in the total study area were observed. Thus, from the families Apiaceae, Cyperaceae, Fagaceae, Orobanchaceae and Ranunculaceae, no species were identified in P2UD (Figure 8b). In this plot, 64 different plant species were recognized.

As was the case in P1UC and P2UD, in P3TD, the botanical families Asteraceae, Fabaceae and Poaceae stand out, with the highest number of identified species (Figure 8c). In this case, the Asteraceae family represented 7 species in WI, 10 in SP1 and 6 in SP2. Regarding the Fabaceae family, the highlight values were observed in SP1, with nine identified species. From the Poaceae family, five species were identified in WI, six in SP1 and seven in SP2. No species were identified in the P3TD plot from the botanical families Cucurbitaceae, Cyperaceae, Fagaceae, Lythraceae, Orobanchaceae and Rubiaceae (Figure 8c). In this plot, 65 different plant species were identified.

The most prominent botanical families in P4TC are Asteraceae, Fabaceae and Poaceae (Figure 8d). The Asteraceae family comprised six species in WI and eight in SP1 and SP2. From the Fabaceae family, seven species were observed in SP1 and only one in WI and SP2. The most significant family was Poaceae, with 6 species identified in WI, 11 in SP1 and 10 in SP2. From the families Apiaceae, Araceae, Cucurbitaceae, Fagaceae, Iridaceae, Juncaceae, Lythraceae, Myrsinaceae, Orobanchaceae and Rubiaceae, no species were identified in P4TC, as can be seen in Figure 8d. In this plot, only 60 different plant species were identified.

#### 3.3.2. Seasonal Bioindicators

Of the 103 plant species observed during the experimental study (51 in WI, 78 in SP1 and 53 in SP2), 18 species can be considered bioindicators (Figure 9). Bioindicators are plant species that are characteristic of a determinate treatment or season of the year [4,36]. Figure 9 represents a diagram of the bioindicator species in each season (WI, SP1 and SP2) according to the ISA application. There were eight bioindicator species in WI and three in SP2, and no significant differences were observed for any species in SP1. In the WI_SP1 combination, there were three bioindicator species, and in the SP1_SP2 combination, there were four bioindicator species (Figure 9).

Figure 10 presents the analysis diagram of the ISA application to verify the existence of bioindicator plants for each treatment (P1UC, P2UD, P3TD and P4TC). As shown in Figure 10, only P4TC had bioindicator species: a total of three bioindicator species. Furthermore, in the P1UC_P2UD combination, there is one bioindicator species. Thus, only four species proved to be bioindicators of the four treatments.

Table 1 refers to the analysis diagram of the ISA application to verify the existence of bioindicator plants for the different combinations between seasons (WI, SP1 and SP2) and treatments (P1UC, P2UD, P3TD and P4TC). In total, 25 bioindicator species were identified for different combinations, as seen in Table 1.

## 4. Discussion

Management of the Montado ecosystems is highly complex, as it comprises several interconnected subsystems that influence each other. The diversity of plant species in the Montado pastures increases the complexity of this ecosystem [4]. Pasture degradation is often associated with a high animal stocking rate. Sometimes, there is confounding between high stocking rates and poor pasture management. Poor management of pastures and high stocking rates can contribute to overgrazing, reduced available biomass and the degradation of pasture and soil [29]. However, other studies show that a high animal stocking rate, per se, is not a factor in soil and pasture degradation, as long as the response capacity of the pasture is taken into account and the regrowth capacity in the more critical seasons is preserved [14,23,29].

### 4.1. Relationship between Climatic Variables, Pasture Development and Grazing Days

The irregularity of the Mediterranean climate influences the germination and growth dynamics of annual pastures’ species. In this study field, precipitation and temperature showed significant climatic variability since 2015, with some very dry autumns delaying the pasture germination and others with large amounts of precipitation [4]. Moreover, the same phenomena occur in the spring: some years have rainy springs and others are very arid [4]. Autumn 2020’s precipitation values did not compromise the germination and development of the pasture since, between September and October, the precipitation value was 148 mm. For the pasture to germinate in autumn, 50 mm of precipitation is required [46]. However, the late of rain in September implied a generalized delay in germination. This lower biomass availability during October led the animals to start grazing only in November, when the average pasture height was around 10 cm. 

The minimum temperatures in December and especially January had a negative effect on the development of the pasture. In some periods of January, a pause in the growth of the pasture was noticeable. This low temperature reduced the use of the pasture in the month of December in deferred grazing systems and the absence of animals during the whole of January. The growth of most of the species that composed this natural pasture, with temperatures as low as 8 to 10 °C, was reduced [47].

In February, the average temperature (11.5 °C), combined with a high value of precipitation (116.5 mm), allowed the regrowth of the pasture and, consequently, grazing in the P2UD and P3TD plots during the last 17 days of the month, extending into the middle of March.

During the experimental period, spring was also quite dry, which may have compromised the length of the vegetative cycle of the pasture.

Although the average temperature (13 °C) in March was favorable to pasture growth, the total precipitation (12 mm) compromised its growth. In addition to this low value of precipitation, the strong wind that occurred on some days also negatively affected plant growth, which interrupted the DG for a few days. Added here is the negative effect of the minimum average temperature in March, which recorded a value of 6.8 °C.

However, the amount of precipitation in April (around 60 mm), with an average temperature of around 15 °C, permitted pasture growth and the lengthening of this vegetative cycle for a few more days. Rainfall in April is significant for the growth and development of the pasture [46]. These low values of precipitation in the spring, together with the increase in temperatures, may have affected the phenological cycle of the different species of the pasture. Moreover, the temperatures in the months of May and June could have enabled the development of the pasture, as well as the extension of the vegetative cycle, were it not for the low precipitation values (7.7 mm and 10 mm, in May and June, respectively) of this period. In any case, the precipitation in April, combined with the spring temperatures, promoted the growth of the pasture, which allowed the grazing in P2UD and P3TD, during May and June. Furthermore, temperatures in May were not very high, thus reducing pasture evapotranspiration. This allowed for the maintenance of soil moisture for a longer time. Grazing days have always depended on pasture growth. In WI, the limiting factor was temperature, while in SP1 and SP2, it was precipitation.

### 4.2. Evolution of the Floristic Composition: Field Observations

The results of this study reflect the effect of differentiated grazing over a period of two years and the application of dolomitic limestone since 2017.

In this study, 103 different plant species were identified, pertaining to 25 botanical families. The botanical families Asteraceae, Fabaceae and Poaceae were the most represented in all the studied plots and in all seasons, although with some variations. However, the families Plantaginaceae and Polygonaceae also had considerable representation. 

According to Serrano et al. [4], the pH in P3TD and P4TC in March 2020 was 5.7, with a small increase compared to October 2015, when the pH value was 5.4. This increase in pH due to the two soil amendments performed in the field may not have been enough to cause significant changes in the PFC. On the other hand, the Mg/Mn ratio also increased [4], which may have benefited the emergence of some plant species, namely legumes. However, Mn toxicity’s influence on plants depends on the species and cultivars [48].

The number of species in the Fabaceae family was always much higher in SP1 than in other seasons. Soil acidity and Mn toxicity harm leguminous plants [48]. However, in this study, DG seems to have exerted a positive effect on legumes since, in P2UD, where no soil amendment was applied, the number and species of legumes in WI, SP1 and SP2 were precisely equal to the plot P3TD, where 2 ton/ha of dolomitic limestone was applied, which did not occur in P1UC. This may be explained by the fact that deferred grazing with a high stocking rate allows grasses to be ingested more because they are more palatable than legumes during the winter, and thus provides more plentiful access to light for legumes. In a similar study, after 3 years of DG, the density of perennial grasses increased to 88%, decreasing the density of annual grasses up to 58%, contributing to increased pasture DM production and improved PFC, soil cover and system resilience [28]. Nevertheless, this same study also reports that DG did not affect the density of legumes. In P4TC, the number of legume species was significantly reduced, with only one in WI and SP1 and seven in SP2. It is likely that this is due to the application of the soil amendment in this plot, and, despite the CG, the stocking rate was low, which led to the substantial initial growth of grasses in the autumn, which tends to shade out the leguminous plants, limiting their growth. When the animal stocking rate is high enough to ingest the produced biomass, the competition for light is reduced, thus allowing the growth of plants of the Fabaceae family [49]. Ferreira et al. [50] reported that the exclusion of grazing had a negative effect on prostrate plants, where some legumes are included. In our study, in P4TC, although there was CG, there were few animals to remove the pasture production, and thus prostrate plants, such as legumes, were affected. This probably occurred due to the lack of light in the lower layers of the pasture. The sample points where the pasture presented lower and more uniform height were also those where the greatest presence of legumes was observed. For example, this effect was observed in P4TC, which can be associated with the animal’s preferred grazing, where the legumes are more competitive for light access (unpublished data). According to Heady [15], when grazing, sheep seek species that are rich in crude protein and have a low content of crude fiber. This selectivity can lead to better animal performance [39]. At an early stage in the growing cycle, sheep do not eat legumes and have a clear preference for grasses and other species. Moreover, in the other plots, leguminous species were identified mainly in the grazing areas preferred by the animals, although, in the plots destined for DG, the selectivity was very low. Nonetheless, grazing with a high stocking rate during the winter enhances most pasture species’ growth, especially ryegrass and red clover [11]. 

### 4.3. Evolution of the Floristic Composition: Field Observations vs. Indicator Species Analysis

In the statistical analyses (ISA), there were no significant differences between plots or seasons for the Fabaceae family. In a study carried out by Nie and Zollinger [28], in which they compared the application of fertilizer and amendments (50 Kg P + 2 ton/ha dolomitic limestone), with no application of fertilizer or amendments, they found that the first treatment contributed to the increase in the density of leguminous plants by 60%, without any effect on other plant families. In a study in New Zealand, in natural pastures, the effect of CG vs. RG was not significant in any species of pasture plant [25]. However, in the same study, in pastures overseeded with five species (*Cirsium vulgare*, *Lolium perenne*, *Plantago lanceolata*, *Rumex obtusifolius* and *Trifolium repens*), seedling density was higher in RG plots when compared to plots with CG. Leguminous plants are directly related to the nutritional quality of the pasture. In this study, *Trifolium repens* was observed during SP1 at many sampling points, in plots with DG. In P4TC, it was only observed in SP1 and at sampling points preferred by the animals. DG with high instantaneous biotic loads appears to be relevant for increasing rangeland biodiversity, increasing desirable plants and reducing undesirable ones. Leguminous plants, especially *Trifolium repens*, are essential in pastures since they provide high-quality food and fix nitrogen in the soil [19]. Furthermore, *Trifolium repens* is quite tolerant to grazing and treading [19], which means that DG does not restrain its development.

In this study, the plant species diversity was higher in P1UC (36 in WI, 58 in SP1 and 37 in SP2). On the contrary, the lowest botanical diversity was observed in P4TC, with only 29, 41 and 29 plant species identified in WI, SP1 and SP2, respectively. For SP1, all species have the same chance of appearing in all treatments in that season. It should be noted that *Lolium perenne* is a SP2 bioindicator species despite being a grass (*Poaceae* family). However, we must point out here that bioindicators can be negative—that is, certain species not being bioindicators can be an advantage for the improvement of PFC. For example, species of the genus *Rumex* were not bioindicators in SP1 or SP2, or in any of the four treatments, which means that they tend to disappear, which is advantageous for sheep production systems, as these plants are unpalatable and have low nutritional value. Regarding DG, in P2UD, 30, 51 and 19 plant species were identified in WI, SP1 and SP2, respectively; in P3TD 31, 47 and 22 species were identified in WI, SP1 and SP2, respectively. At the end of the vegetative cycle, the botanical diversity was higher in the CG plots than in the DG plots. Similarly, the same happened in the studies of Edwards et al. [25] and Marley et al. [37], where the species diversity was higher with CG than with RG. *Diplotaxis catholica* is considered a weed plant in Mediterranean pastures, and is only consumed by grazing animals in the first phenological stages, always before maturation and, above all, if the instantaneous animal stocking rate is high. In this study, this species was no longer observed in SP2, except in P2UD. The presence of this species may indicate that the high animal stocking rate and the consequent reduction in selectivity led to its total consumption during SP1. *Echium plantagineum* was present in all plots and in all seasons. Another species that was also present in all plots in WI and SP1 was *E**rodium cicutarium*. However, in SP2, it was only identified in P4TC, probably because this species, at the end of the vegetative cycle, has sharp structures (stubble) that prevent animals from eating it, which may have led to it not being ingested. In the other plots, this did not occur because the animals ingested the plants before this phenological stage. Sometimes, the dominant plants in a pasture are unwanted plants with reduced palatability and nutritional value for animals. As they are not consumed or preferred, they become dominant, leading to pasture degradation. Grazing with a high stocking rate during winter boosts all pasture species’ growth, especially ryegrass and red clover [11].

### 4.4. Evolution of the Floristic Composition: Effects of Different Gazing Management

Grazing management is essential for maintaining functional ecosystems and contributes to the biodiversity of species. A study carried out by Mendes et al. [49] in an area dominated by *Cistus ladanifer* L. shrubs, with five types of management—abandonment; initial cutting and grazing with 2 to 3 normal heads/ha; cutting every two years; fire after five years of abandonment; soil mobilization (and abandonment)—showed that only cutting and grazing led to a significant reduction in shrubs and increased herbaceous species, especially from the Poaceae and Fabaceae families. Moreover, Ferreira et al. [50] reported that excluding grazing harms species diversity.

In P2UD and P3TD, the number of grazing days and the interval between each grazing period depended on the height and quality of the pasture. Ferreira et al. [50] state that the interval between grazing periods depends on the place and the season of the year, i.e., it depends on the conditions of the pasture. 

DG, in which grazing periods are defined according to pasture conditions, is the most effective method for increasing perennial grasses and reducing annual ones, which can help to improve the PFC more quickly [28]. Mendes et al. [49] reported that proper grazing management tends to decrease invasive shrubs (*Cistus ladanifer*) and increase the Poacea and Fabacea families, especially *Poetea bulbosae*, *Poa bulbosa* L., *Trifolium subterraneum* L., *Erodium botrys* (Cav.) Bertol., *Trifolium glomeratum* and *Trifolium tomentosum* L. Seven months without grazing led to a 72 to 87% reduction in the density of grasses, clovers and other species [51]. 

The plant species that make up the pasture can affect the feeding efficiency of the animals [29], as well as the grazing system. Moreover, the plant families determine the feeding quality of the pasture, and according to [21], legumes generally have better nutritional value than grasses: the more legumes in the pasture, the greater will be its quality [52]. 

The sheep’s body condition was not affected during the experimental period. During the production cycle, monitoring of the energy balance and quantification of the animal body’s reserve changes are essential and were performed in the field by estimating the body condition (BCS) and its variations [41]. This method evaluates the fat tissue thickness and the muscle on the waist and spine. The BCS is described as the ratio of total fat and other tissues on a live animal, and it is crucial to obtain the desired performance in certain physiological states in extensive sheep systems. There can be variable scores within different genotypes and physiological statuses of ewes (Biçer, 1991) cited by [53].

Sometimes, differences are observed in the performance of animals in pastures where there are only grasses, which is due to different proportions of leaves, stems, seeds and/or inflorescences, which vary between grass species [21].

## 5. Conclusions

Extensive livestock production systems, based on rainfed pastures under the Montado, are based on high complexity, resulting from the interactions between soil, pasture, trees and animals, together with precipitation and temperature, throughout the year. Despite pasture being the cheapest food for ruminants, its production and improvement in terms of quality and nutritional value are not always easy to implement in a complex production system such as the Montado. An essential component still poorly studied is the PFC and the interactions between it, the animals, the type of grazing and soil properties (namely acidity and Mn toxicity). The PFC of the pasture is responsible, above all, for its quality.

Statistically, there were no significant differences in the probability of occurrence of certain species in P1UC, P2UD and P3TD. However, in P4TC, three plant species were identified as bioindicators of this treatment (*Crassula tillaea* Lest.-Garl., *Poa bulbosa* and *Ranuncullus ollissiponensis* Pers.). For each season of the year and for their combinations, several bioindicator plants were identified. The most representative botanical families in all study plots were Asteraceae, Fabaceae, Poaceae, Plantaginaceae and Polygonaceae. The Fabacea family was widely present in SP1. 

DG appears to be beneficial for eliminating undesirable species and the consequent increase in desirable species, and from the sheep’s point of view, there seems to be no disadvantage as the nutritional value tends to be higher. 

The application of dolomitic limestone combined with CG proved to be inefficient in increasing the biodiversity of the pasture, as well as in increasing the number of prostrate-sized plant species, such as those belonging to the genus Trifolium. The sheep’s body condition during the experimental period did not differ among treatments. 

A better understanding of the effects of sheep grazing, stocking rates and dolomitic limestone application on PFC can have a strong impact on the improvement of extensive livestock production systems in the Mediterranean region. Thus, this work can significantly contribute to more informed decision-making among farmers, ensuring the efficiency and the sustainability of the Montado ecosystem.

## Figures and Tables

**Figure 1 animals-12-02506-f001:**
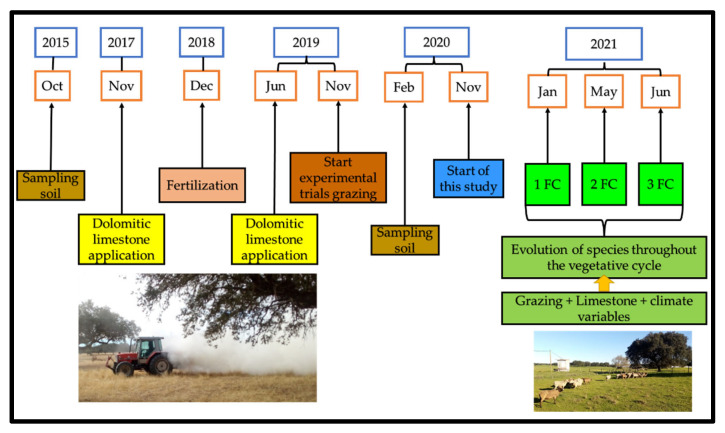
Chronological diagram of the study, with its framework in the research project of soil and pasture monitoring, in the Montado ecosystem (FC—characterization of floristic composition).

**Figure 2 animals-12-02506-f002:**
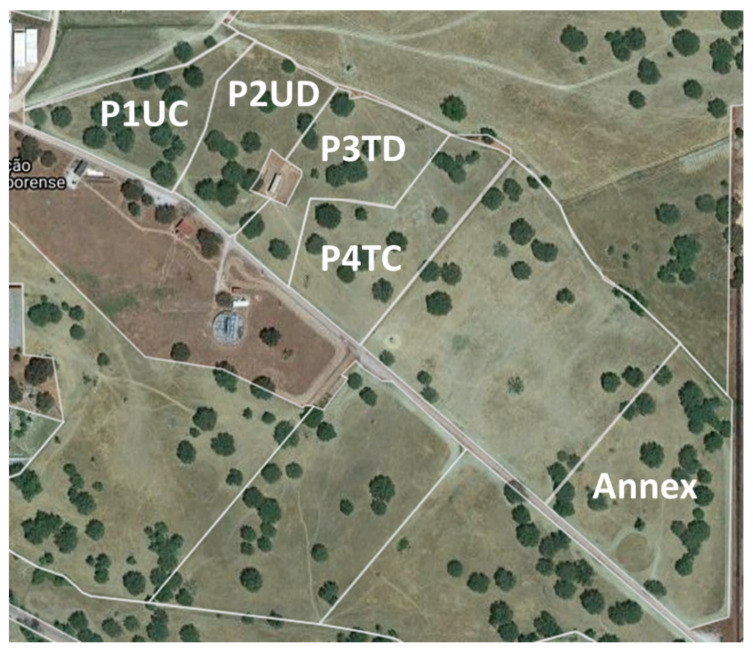
Four areas corresponding to four treatments are represented by U—without dolomitic limestone, T—with dolomitic limestone, C—continuous grazing, D—deferred grazing within the study area and remaining annex area.

**Figure 3 animals-12-02506-f003:**
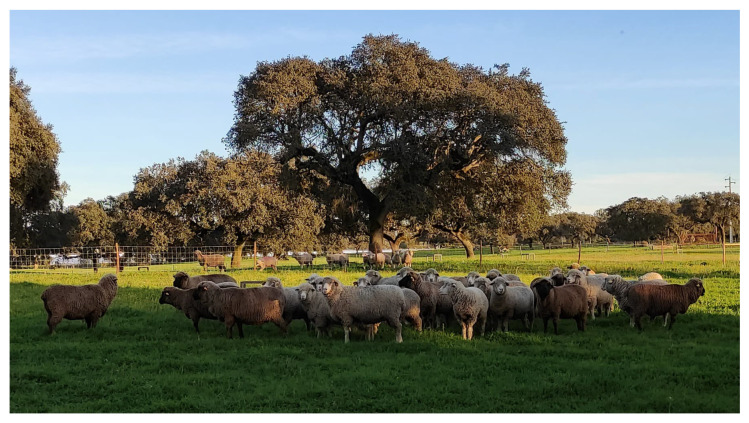
White Merino and Black Merino sheep in a plot of the study area.

**Figure 4 animals-12-02506-f004:**
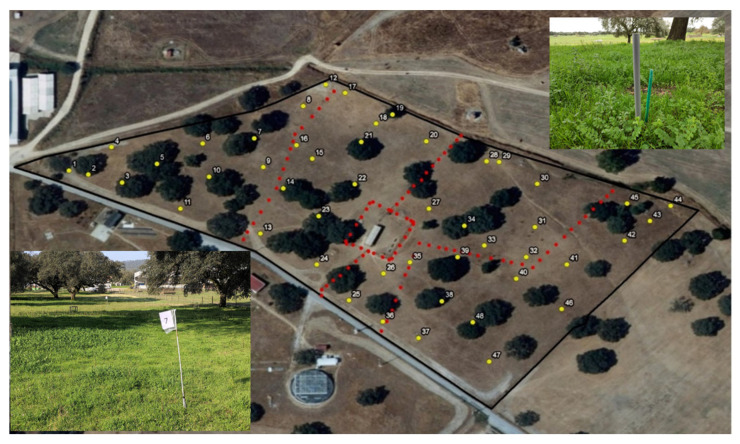
Representative sampling points of different pasture plant communities and sample point marking example.

**Figure 5 animals-12-02506-f005:**
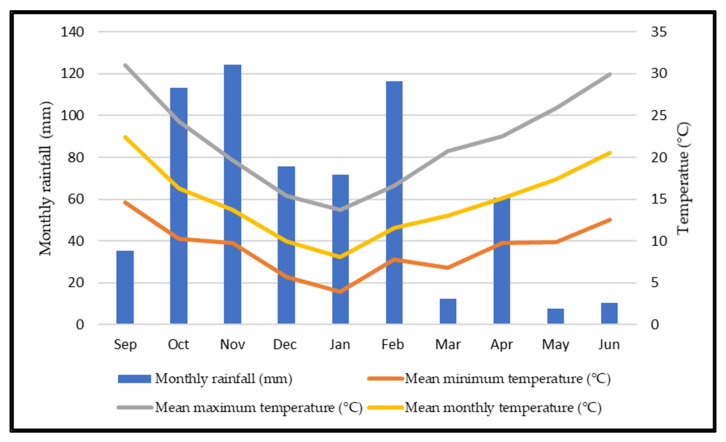
Thermopluviometric graph for Mitra station (Évora) between September 2020 and June 2021.

**Figure 6 animals-12-02506-f006:**
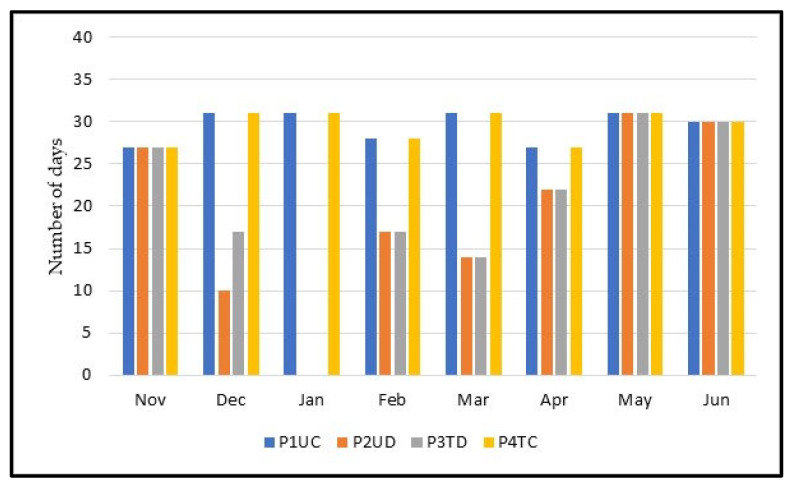
Number of grazing days in each treatment, per month.

**Figure 7 animals-12-02506-f007:**
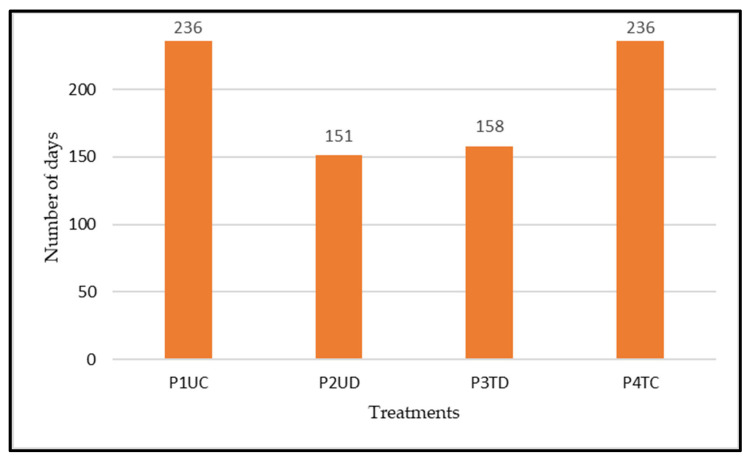
Total grazing days in each treatment.

**Figure 8 animals-12-02506-f008:**
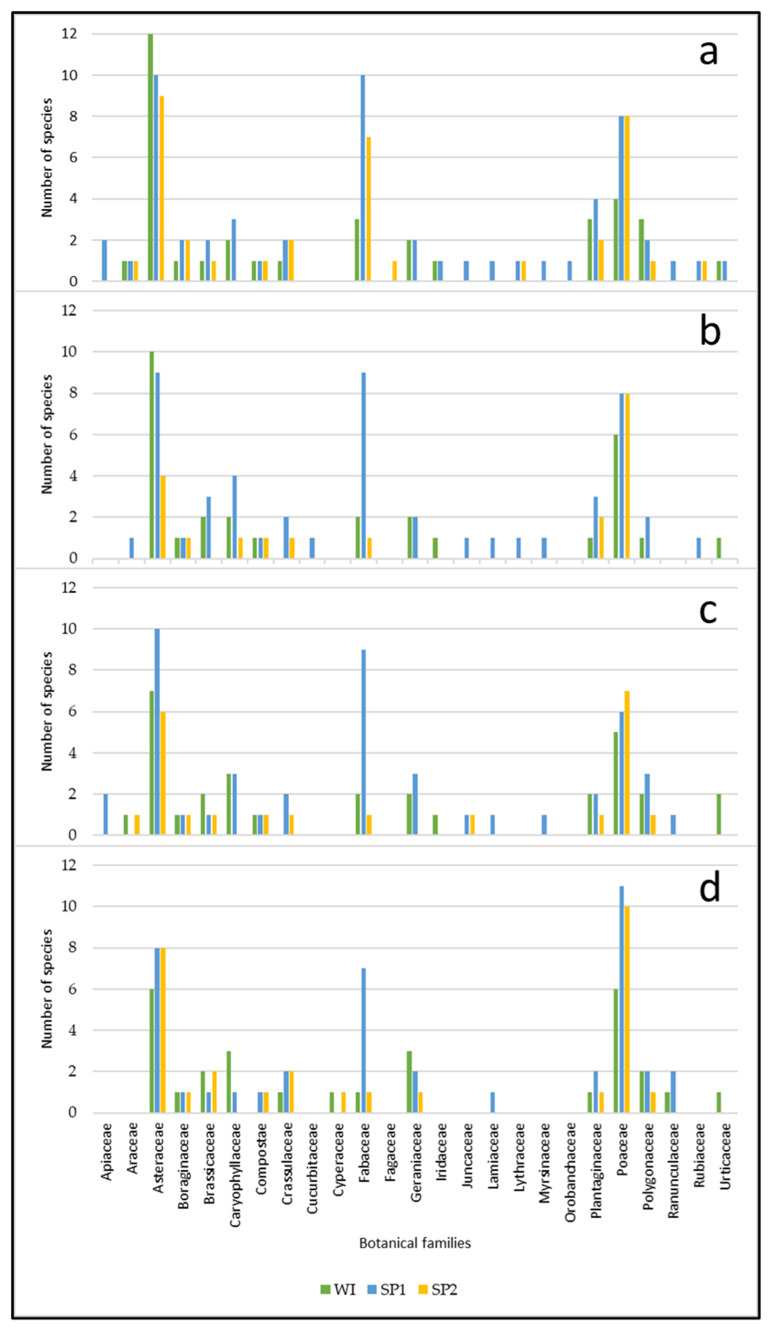
Number of plant species per family for P1UC (**a**), P2UD (**b**), P3TD (**c**) and P4TC (**d**) in WI, SP1, SP2.

**Figure 9 animals-12-02506-f009:**
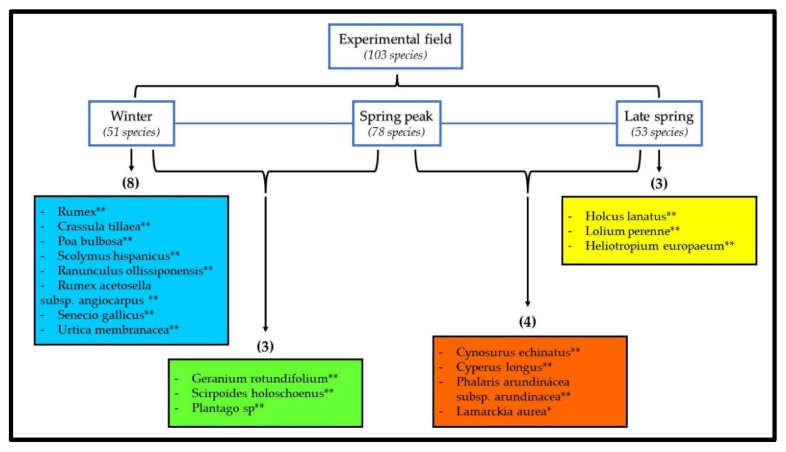
Representative diagram of the analysis of indicator species (ISA), for each season of the year (WI, SP1 and SP2), and their combinations. **—Probability < 0.01; *—Probability < 0.05.

**Figure 10 animals-12-02506-f010:**
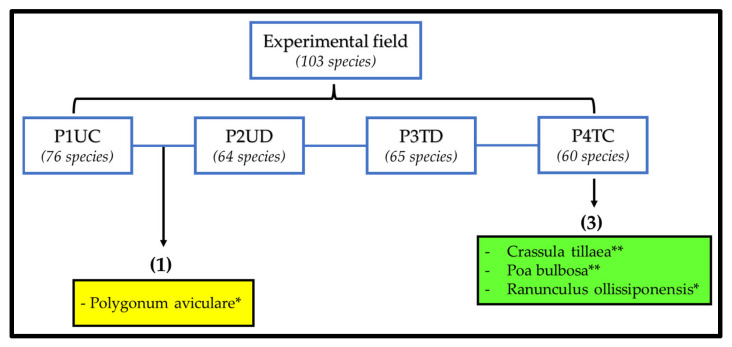
Representative diagram of the analysis of indicator species (ISA), for each plot (P1UC, P2UD, P3TD and P4TC) and their combinations. **—Probability < 0.01; *—Probability < 0.05.

**Table 1 animals-12-02506-t001:** Analysis of indicator species (ISA) for bioindicator species for the combinations between season (WI, SP1 and SP2) and treatment (P1UC, P2UD, P3TD and P4TC).

Combinations: Seasons and Plots	Species	*p*-Value
WI_P1UC; WI_P2UD; WI_P3TD	*Urtica membranacea*	0.005 **
WI_P1UC; WI_P2UD; WI_P3TD; WI_P4TC	*Rumex* sp.	0.005 **
	*Crassula tillaea*	0.005 **
	*Poa bulbosa*	0.005 **
	*Ranunculus ollissiponensis*	0.005 **
	*Silene gallica*	0.005 **
WI_P1UC; WI_P2UD; WI_P3TD; WI_P4TC; SP1_P3TD	*Plantago* sp.	0.005 **
WI_P1UC; WI_P2UD; WI_P3TD; WI_P4TC; SP1_P1UC; SP1_P3TD	*Scolymus hispanicus*	0.005 **
WI_P1UC; WI_P2UD; WI_P3TD; WI_P4TC; SP1_P2UD; SP1_P3TD; SP1_P4TC	*Rumex acetosella* subsp. *angiocarpus*	0.005 **
SP2_P1UC; SP2_P2UD; SP2_P3TD; SP2_P4TC	*Holcus lanatus*	0.005 **
SP2_P1UC; SP2_P2UD; SP2_P3TD; SP2_P4TC; SP1_P4TC	*Cyperus longus*	0.005 **
	*Phalaris arundinacea* subsp. *arundinacea*	0.005 **
	*Lolium perenne*	0.005 **
SP1_P1UC; SP1_P2UD; SP1_P4TC; SP2_P1UC; SP2_P2UD; SP2_P3TD; SP2_P4TC	*Cynosurus echinatus*	0.005 **
WI_P1UC; WI_P2UD; WI_P3TD; WI_P4TC; SP1_P1UC; SP1_P2UD; SP1_P3TD; SP1_P4TC	*Geranium rotundifolium*	0.005 **
	*Scirpoides holoschoenus*	0.005 **
	*Leontodon tuberosus*	0.005 **
WI_P1UC; WI_P2UD; SP1_P1UC; SP1_P4TC; SP2_P1UC; SP2_P2UD; SP2_P3TD; SP2_P4TC	*Heliotropium europaeum*	0.035 *
WI_P4TC; SP1_P1UC; SP1_P2UD; SP1_P3TD; SP1_P4TC; SP2_P1UC; SP2_P2UD; SP2_P3TD; SP2_P4TC	*Tolpis barbata*	0.005 **
WI_P2UD; WI_P4TC; SP1_P1UC; SP1_P2UD; SP1_P3TD; SP1_P4TC; SP2_P1UC; SP2_P2UD; SP2_P3TD; SP2_P4TC	*Orobanche* sp.	0.01 **
WI_P1UC; WI_P2UD; WI_P3TD; WI_P4TC; SP1_P1UC; SP1_P2UD; SP1_P3TD; SP1_P4TC; SP2_P1UC; SP2_P3TD; SP2_P4TC	*Stellaria media*	0.005 **
	*Sonchus oleraceus*	0.02 *
WI_P1UC; WI_P3TD; WI_P4TC; SP1_P1UC; SP1_P2UD; SP1_P3TD; SP1_P4TC; SP2_P1UC; SP2_P2UD; SP2_P3TD; SP2_P4TC	*Quercus rotundifolia*	0.005 **
WI_P2UD; WI_P3TD; WI_P4TC; SP1_P1UC; SP1_P2UD; SP1_P3TD; SP1_P4TC; SP2_P1UC; SP2_P2UD; SP2_P3TD; SP2_P4TC	*Bromus tectorum*	0.005 **
	*Lolium rigidum* subsp. *rigidum*	0.005 **

**—Probability < 0.01; *—Probability < 0.05.

## Data Availability

Not applicable.

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
