# Peer review of "Effect of Sheep Grazing, Stocking Rates and Dolomitic Limestone Application on the Floristic Composition of a Permanent Dryland Pasture, in the Montado Agroforestry System of Southern Portugal"

_animals, 2022, doi:10.3390/ani12192506_

Round 1

Reviewer 1 Report

Dear Authors,

Your manuscript covers some significant aspects of grassland ecology and management. Below are a few suggestions to improve the soundness of the manuscript. 

(1) In the introductory section (Lines 77 - 80), reexamined how the two terms grazing systems and management are used. Grazing systems integrate components of animal, plant, soil, environment, and management.  While grazing management involves the manipulation of the soil-plant-animal complex to achieve the desired results (e.g., Extensive vs Intensive). Thus continuous grazing, rotational grazing, and deferred grazing are Grazing Methods/ Stocking Methods. In other words, they are a component of the grazing systems.    

(2) Under the Materials and Methods section, (line 190) was there replications of treatments in this study? If not how was this accounted for?

(3) Line 212, the way how the reference is cited makes the sentence reading incomplete.  

(4) Lines 218 - 219, 'Pasture height .... something is missing to make this sentence complete. possibly it should be 'Pasture height measurements.'

(5) Lines 475 - 477, need restructuring for smooth reading. Especially the way how Serano et al. [4] was cited. 

Overall, I would have preferred to see a more robust method used to quantify floristic composition in this study. 

Author Response

Dear reviewer,

Thank you very much for your suggestions, questions and comments.

In general, your suggestions were accepted. Your contributions have improved the article.

In the attachment we send a document with some answers and justifications. Changes can be found in the article, highlighted in yellow. What is highlighted in yellow, but cut off, is to be deleted.

In the pdf document are all the changes, with what was added and what was cut. In the word document is only the content that is to stay. Changes are highlighted in yellow.

Yours sincerely,

Best regards

Emanuel Carreira

Reviewer 2 Report

Dear authors...

my comments regarding your article can be seen in the attached draft, highlighted in yellow

Author Response

Dear reviewer,

Thank you very much for your suggestions, questions and comments.

In general, your suggestions were accepted. Your contributions have improved the article.

In the attachment we send a document with some answers and justifications. A pdf document will be submitted with all changes made highlighted in yellow. Contents to be deleted are highlighted in yellow, but cut. Afterwards, a word document will be submitted with only the changes that must remain in the article, highlighted in yellow.

Yours sincerely,

Best regards

Emanuel Carreira

Author Response

(The authors gave the same response as above.)

Reviewer 4 Report

Dear colleagues,

since I had some Adobe problems, some remarks were not inserted in the document:

L 2-3: Effect of animal grazing, stoking rates and dolomitic limestone application on the floristic composition of a permanent dryland pasture, in the Montado

Remark 1: Effect of distinct sheep grazing regimes and limestone applications on the floristic composition of an arid Mediterranean pasture in Portugal

Remark 2: „Montado“ is not known (enough) to appear in a title that shall inform a wide public instantly. “Arid” – would that be a better option instead of “permanent dryland”?

L 21: to evaluate the effect of the application of dolomitic lime to the soil and, of different types of grazing

Remark 3: Order of variables is different from that in the title. Please change.

L22: The results show that climatic variables have a very pronounced effect on pasture composition

Remark 4: According to L. 21, you did yet not aim to investigate the effect of those variables, therefore, you must not mention them in the summary.

L 23: The type of grazing also influences the PFC, which may be positively or negatively impacted, depending on the adopted system

Remark 5: That implies nothing concrete. Please, specify the impact of grazing in few but informative words.

L 26: which allow an increase the efficiency of animal production systems and conservation of the Montado.

Remark 6: Since “production” and “conservation” most often is a contradiction, I would suggest to formulate as follows (note the grammar): This allows not only an increase of animal production but also improves conservation strategies in the Montado.

L30: which affect

Remark 7: affects

Remark 8: General failures

1.       Not quantitative measurements in plant monitoring

2.       No life form as plant traits or phenological appearance (although there are trees on the pastures, there are no data on shade tolerance or shade profit of herbaceous species)

3.       No data whether the indicator plant species were grazed or not.

Author Response

Dear reviewer,

Thank you very much for your suggestions, questions and comments.

In general, your suggestions were accepted. Your contributions have improved the article.

We have to thank him for the extraordinary work, because his contribution definitely improves the article.

In the attachment we send a document with some answers and justifications. A pdf document will be submitted with all changes made highlighted in yellow. Contents to be deleted are highlighted in yellow, but cut. Afterwards, a word document will be submitted with only the changes that must remain in the article, highlighted in yellow.

Yours sincerely,

Best regards

Emanuel Carreira

Reviewer 5 Report

The argument deserves to be published on Animals. However there are several superficilities about the botanical part that have to be solved before that the Ms can be accepted for publication. Citations of authors' names are full of formatting errors. These gross errors are not acceptable even to students. From this I understand that the authors are mostly involved in the study of animals and so I wonder how the plant species were identified? What floras were used? Have herbaria been consulted? I believe in the good faith of the authors but all these things must be specified. Perhaps the collaboration of a botanist could benefit this article. Another 

Author Response

Dear reviewer,

Thank you very much for your suggestions, questions and comments.

Your suggestions were accepted. Your contributions have improved the article.

As for the questions raised by the reviewer, regarding the identification of species, they were answered in the attached document with the responses to the reviewer. However, we emphasize that co-author Carlos Pinto Gomes is Associate Professor with Aggregation in the Department of Landscape, Environment and Planning, at the University of Évora. In addition, he is Researcher at the MED-Mediterranean Institute for Agriculture, Environment and Development, Research Institute and CHANGE—Global Change and Sustainability Institute, Institute for Advanced Studies and Research University of Évora, where he works as a botanist in the identification of botanical species. Links to sites where you can see the publications of this co-author who is part of this research team are sent in the attached document.

In the attachment we send a document with some answers and justifications. A pdf document will be submitted with all changes made highlighted in blue. Contents to be deleted are highlighted in blue, but cut. Afterwards, the word document will be submitted with only the changes that must remain in the article, highlighted in blue.

In addition, parts of the text will also appear highlighted in yellow (first review round) and green (second review round) in the documents.

Thank you so much!

Yours sincerely,

Best regards

Emanuel Carreira

Round 2

Reviewer 1 Report

Dear Authors,

I am satisfied with the changes you have made to the manuscript. 

Author Response

Dear reviewer,

Thank you very much for all the comments and for all the questions.

Your suggestions/corrections improve the final document.

All text has been proofread by an English expert, as suggested by you.

Thank you so much!
Best regards
Emanuel Carreira

Reviewer 4 Report

I went through the revised comments. Unfortunately, I was not satisfied. This I wrote into the document: " I am still missing Line 21. I wrote: "L22: The results show that climatic variables have a very pronounced effect on pasture composition According to L. 21, you did yet not aim to investigate the effect of those variables, therefore, you must not mention them in the summary." I am not sure why you did not comment Line 21, however, since that comment is missing I expect further comments missing, too. Please go through the whole manuscript snd check whether all comments of mine have been recognized by you. May I give you an advice? You could present a table, one column the original sentence, another column my comment, a third column your comment, a fourth column the revised sentence."

Author Response

Dear reviewer,

Thank you very much for all the comments and for all the questions.

We are very grateful for all the thorough work you have done with our manuscript. Your suggestions/corrections greatly improve the final document.

We apologize once again for missing some responses to your comments and suggestions. This time, there were 6 questions/comments that were not answered by mistake. We think that now everything is answered and justified. These missing answers are in an attached document, like the other document sent in the first review phase.

In the attachment we send a document with answers and justifications. A pdf document will be submitted with all changes made in the first review round are highlighted in yellow, and changes made in this second round are highlighted in green. Contents to be deleted are highlighted in green, but cut. Afterwards, a word document will be submitted with only the changes that must remain in the article, highlighted in yellow (first review round) and in green (second review round).

If any clarification or answer is still missing, we would appreciate you telling us clearly what is missing. Please point out if some comments or questions may still have been ignored or not answered correctly.

All text has been proofread by an English expert.

The authors are very grateful for all your work that improves the quality of our manuscript.

Yours sincerely,

Best regards

Emanuel Carreira

Round 3

Reviewer 4 Report

I already commented evrything in the draft.

Author Response

Dear reviewer,

Thank you very much for all the work you are doing with our article.

Thank you very much for your suggestions, questions and comments.

Your contributions have improved the article.

In the attachment we send a document with some answers and justifications. A pdf document will be submitted with all changes made highlighted in blue. Contents to be deleted are highlighted in blue, but cut. Afterwards, the word document will be submitted with only the changes that must remain in the article, highlighted in blue.

In addition, parts of the text will also appear highlighted in yellow (first review round) and green (second review round) in the documents.

Thank you so much!

Yours sincerely,

Best regards

Emanuel Carreira
